**Subject Category:**
Biology (whole organism)

biophysics/developmental biology/evolution

*Myxococcus xanthus*, phenotypic plasticity, multicellularity, reaction norm, physical forces in development

**Authors for correspondence:**
Ana E. Escalante
e-mail: aescalante@iecologia.unam.mx
Mariana Benítez
e-mail: mbenitez@iecologia.unam.mx

# Plastic multicellular development of *Myxococcus xanthus*: genotype–environment interactions in a physical gradient

Natsuko Rivera-Yoshida[1,2,3], Alejandro V. Arzola[4], Juan A. Arias Del Angel[1,2,3], Alessio Franci[5], Michael Travisano[6], Ana E. Escalante[1] and Mariana Benítez[1,2]

[1]Laboratorio Nacional de Ciencias de la Sostenibilidad (LANCIS), Instituto de Ecología, Universidad Nacional Autónoma de México, Mexico City, Mexico
[2]Centro de Ciencias de la Complejidad, Universidad Nacional Autónoma de México, Mexico City, Mexico
[3]Programa de Doctorado en Ciencias Biomédicas, Universidad Nacional Autónoma de México, Mexico
[4]Instituto de Física, Universidad Nacional Autónoma de México, Apdo Postal 20-364, 01000 Cd de México, Mexico
[5]Facultad de Ciencias, Universidad Nacional Autonóma de México, Mexico
[6]Department of Ecology, Evolution and Behavior, University of Minnesota, Saint Paul, MN, USA

 NR-Y, 0000-0001-5721-0621; AVA, 0000-0002-4860-6330; AF, 0000-0002-3911-625X; MT, 0000-0001-8168-0842; AEE, 0000-0001-8147-4598; MB, 0000-0002-4901-2833

In order to investigate the contribution of the physical environment to variation in multicellular development of *Myxococcus xanthus*, phenotypes developed by different genotypes in a gradient of substrate stiffness conditions were quantitatively characterized. Statistical analysis showed that plastic phenotypes result from the genotype, the substrate conditions and the interaction between them. Also, phenotypes were expressed in two distinguishable scales, the individual and the population levels, and the interaction with the environment showed scale and trait specificity. Overall, our results highlight the constructive role of the physical context in the development of microbial multicellularity, with both ecological and evolutionary implications.

# 1. Introduction

Developmental patterns, and phenotype in general, have often been considered as univocal outcomes of the genome. Thus, research has focused—conceptually and experimentally—on studying genetic mechanisms in invariable environmental conditions. However, phenotype has been shown to involve complex, bidirectional interactions between organisms and their environment. For example, light, moisture and nutrient availability in plants, as well as temperature, population density and predator presence in animals, partially determine developmental trajectories [1–6]. The phenotypic repertoire of a given genotype across a range of environmental conditions is called reaction norm [7] and its characterization constitutes the most common approximation to the study of organism–environment interactions. Since development occurs at changing and often organism-modified environmental conditions, increased survival, adaptation and phenotypic innovation can result from flexible phenotypic responses (phenotypic plasticity). Plasticity itself contributes as a cause and not just as a consequence of development and phenotypic transitions in evolution [1,5,8–10].

The origin of multicellularity is a major evolutionary transition in organismal development. Plant and animal models have guided research in organism–environment interactions as well as multicellular development and evolution. However, microorganisms are largely missing in these integrative efforts, despite their ubiquity across environments and the fact that some of them can develop stereotypical multicellular structures. Microbial groups can provide new insights regarding the contribution of the organism–environment interactions to the development and evolution of multicellularity since they develop in a scale and environment similar to those in which multicellularity might have emerged [11]. Importantly, physical processes associated with environments at the microscale happen to be relevant as both driving and restrictive forces in the development of microbial multicellular phenotypes [12–14]. These forces and their effect on living matter vary across scales, leading to reconsideration of the meaning of environment and phenotype at different organization levels.

*Myxococcus xanthus* is a cosmopolitan soil bacterium that has a multicellular life history and moves by gliding across surfaces. In nutrient-rich substrates, cells divide and swarm outwards expanding the colony radially. When nutrients become scarce, *M. xanthus* cells come together and develop into three-dimensional multicellular structures called fruiting bodies (FBs) that contain differentiated cells (spores) [15]. The genetics underlying this process have been studied intensively [15,16]. In addition to the multicellular organization at the level of a single FB, the development of *M. xanthus* involves a stereotypical, but largely unexplored, spatial arrangement of FBs [17]. Given their characteristic gliding motility, cell-to-substrate interaction is an important aspect of organism–environment interaction in myxobacteria and is susceptible to variation by mechanical properties of the substrate [18–20]. We therefore hypothesize that changes in the substrate stiffness may affect velocity and aggregate formation, uncovering phenotypic plasticity in different scales and traits of the multicellular structures of *M. xanthus*.

In the present study, we determine reaction norms of different *M. xanthus* genotypes in multicellular development under varying substrate stiffness environments. Our results show that phenotypic plasticity for multicellularity in *M. xanthus* involves multiple levels of biological organization: single FBs and FB groups.

# 2. Methods

## 2.1. Strains, growth and developmental conditions

To evaluate the contribution of the genotypic and environmental context to the phenotypes, five genotypes (strains) were assayed for development over five different substrate stiffness conditions, which were modified by varying agar concentrations (parental strain: DZF1; in-frame deletion mutants: $\Delta mkapC$, $\Delta mkapA/\Delta mkapC$, $\Delta mkapA$, $\Delta pktC2/\Delta pktD1$, kindly provided by S. Inouye; agar concentrations: 0.5%, 1.0%, 1.5%, 2.0%, 2.5%) [21]. DZF1 is a standard laboratory *M. xanthus* strain. $\Delta mkapC$, $\Delta mkapA/\Delta mkapC$, $\Delta mkapA$ and $\Delta pktC2/\Delta pktD1$ are DZF1-derived in-frame deletion mutants of the network of PSTKs and the associated scaffold proteins Mkaps, which participate during the development of FBs [16,22]. An important feature of these mutants is that they do not arrest development and thus allow tracking of multicellular phenotypic changes at this scale. Previous studies suggest that deletion of *pktC2*, *pktD1* and other components of this network (*pktA2*, *pktD9*) impacts development and FB formation [17,23–26]. MkapA and MkapC are scaffold proteins in this network, but there are not previous reports regarding the phenotypic consequences of their deletion.

Following the protocol described in Yang & Higgs [15], strains were taken from frozen stocks by spotting 50 μl of each onto a CYE agar plate and incubated at 32°C for 2 days. Cells from the resulting

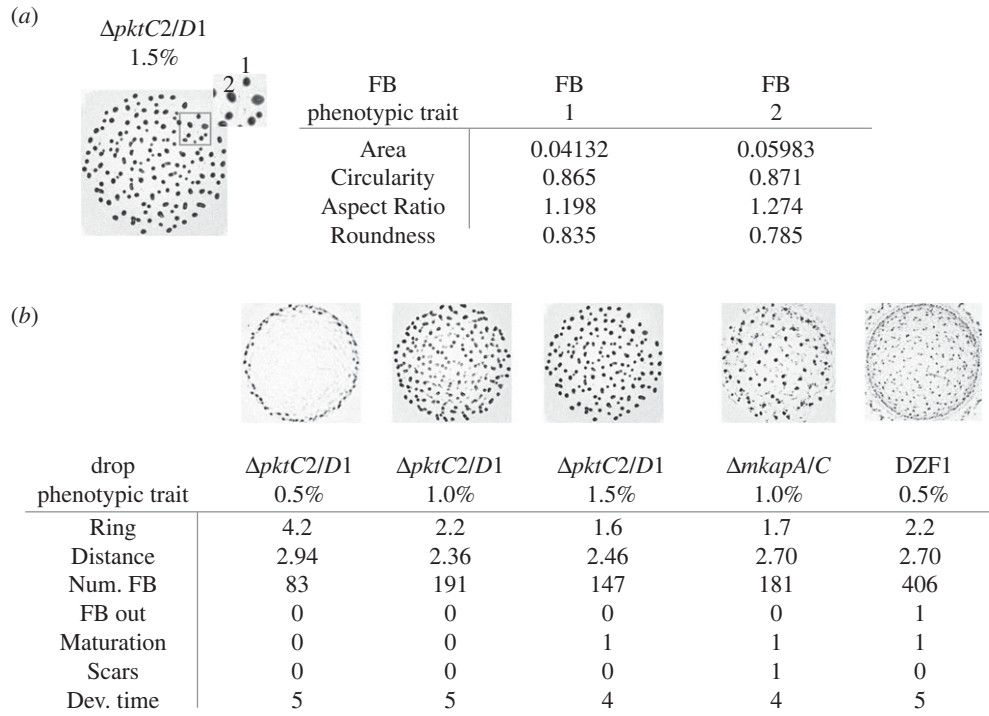

| FB phenotypic trait | FB 1 | FB 2 |
|---|---|---|
| Area | 0.04132 | 0.05983 |
| Circularity | 0.865 | 0.871 |
| Aspect Ratio | 1.198 | 1.274 |
| Roundness | 0.835 | 0.785 |

| drop phenotypic trait | ΔpktC2/D1 0.5% | ΔpktC2/D1 1.0% | ΔpktC2/D1 1.5% | ΔmkapA/C 1.0% | DZF1 0.5% |
|---|---|---|---|---|---|
| Ring | 4.2 | 2.2 | 1.6 | 1.7 | 2.2 |
| Distance | 2.94 | 2.36 | 2.46 | 2.70 | 2.70 |
| Num. FB | 83 | 191 | 147 | 181 | 406 |
| FB out | 0 | 0 | 0 | 0 | 1 |
| Maturation | 0 | 0 | 1 | 1 | 1 |
| Scars | 0 | 0 | 0 | 1 | 0 |
| Dev. time | 5 | 5 | 4 | 4 | 5 |

**Figure 1.** Micrographs to exemplify the quantification of *M. xanthus* phenotypic traits at (*a*) FB scale (Area, Circularity, Aspect Ratio, Roundness), and (*b*) population scale (Ring, Distance, Num. FB, FB out, Maturation, Scars, Dev. time). Black dots correspond to developed FBs at 96 h after starvation. Phenotypic trait units are provided in electronic supplementary material, table S1.

colonies were transferred to 25 ml of CYE liquid medium and incubated at 32°C, shaking at 250 r.p.m. overnight. Culture dilutions of each strain were grown from 0.2 OD550 until they reached 0.7 OD550 (nutrient-rich liquid culture). Prior to the bacteria development assays, cells were harvested by spinning them at 8000 r.p.m. for 5 min. The resulting pellet was washed twice with TPM solution and resuspended in 1/10th of the original volume. Fifteen microlitres were spotted onto the different agar concentration TPM plates. TPM plates were prepared by filling each with 30 ml TPM/agar media, at the appropriate agar concentration, and storing them overnight at 32°C before use. After the spots dried, the plates were incubated at 32°C for 96 h, until aggregates fully developed into matured FBs, recognized by their invariant shape, size and complete darkening [17]. In order to measure the motility of each strain, 15 µl of nutrient-rich liquid culture (CYE, described above) was spotted onto the different agar concentration CYE plates. After the spots dried, plates were incubated at 32° for 96 h. Each strain/substrate condition of TPM and CYE plates were conducted in triplicate for statistical support.

## 2.2. Measurement of phenotypic traits

Micrographs of the resulting FB development on nutrient-poor substrates, or growth on nutrient-rich substrate, were taken at 370.8 pixels mm$^{-1}$ using a LEICA m50 stereomicroscope with an ACHRO 0.63× objective lens and a Canon-EOS Rebel T3i camera. For image processing, micrographs were binarized into black/white images and phenotypic traits were measured using FIJI (ImageJ) software v. 2.0.0 [27]. For each FB image, phenotypic traits were classified and quantified at two scales: individual and population level. Single FB scale traits included Area, and Circularity, Aspect Ratio and Roundness as shape descriptors. Population scale traits, from all the FBs resulting from a single drop, included ring formation at the edge of the drop (Ring), standard distance between FBs (Distance; note that this variable might change in unexpected directions when the FBs exhibit an uneven distribution), number of FBs (Num. FB), maturation of FBs considered as complete darkening of clearly defined aggregates (Maturation), presence of vestiges of not clearly defined aggregates (Scars), position in the sequence of maturation speed, being 1 the fastest agar % and 5 the slowest agar % for each genotype (Dev. time), and mature FBs beyond the edge of the drop (FB out). Outer FBs are formed during development from cells that migrated beyond the initial drop edge. In order to analyse the images systematically, we cropped the FBs placed outside the drop, but recorded their presence under the categorical variable 'FB out' (figure 1; electronic supplementary material, table S1)

**Table 1.** PERMANOVA analysis to test the contribution of genotype, agar concentration and their interaction on *M. xanthus* phenotypes.

|  | d.f. | sums of squares | mean of squares | pseudo-$F$ | $R^2$ | $p$-value |
|---|---|---|---|---|---|---|
| genotype | 4 | 35.51 | 8.87 | 4321.8 | 0.571 | 0.000999 |
| agar | 4 | 10.35 | 2.58 | 1259.9 | 0.166 | 0.000999 |
| genotype – agar | 16 | 11.48 | 0.71 | 349.6 | 0.185 | 0.000999 |
| residuals | 2305 | 4.73 | 0.002 |  | 0.076 |  |
| total | 2329 | 62.08 |  |  | 1.000 |  |

[28]. The edge of the drop was defined at time 0 h, when it is clearly seen, but it did not expand or change its position during development on nutrient-poor media.

For assays of growth on nutrient-rich substrate, the diameter of colonies was measured in micrographs taken from 0 to 96 h, every 24 h. We calculated the growth rate as the percentage that the colony diameter increased over 96 h. For this study, this rate was used as a proxy for motility. These measurements were performed using FIJI (ImageJ) software v. 2.0.0 [27].

## 2.3. Data analysis

In order to test contributions of genotype, substrate and their interaction, a PERMANOVA analysis of a representative sample was performed using the *adonis* function in the R package *vegan* (10% of the total data, $N = 2400$) (table 1) [29]. This is a multivariate test suitable to distinguish sources of variation in non-parametric datasets. To investigate the phenotypic differences in micrographs of figure 2, phenotypic traits were grouped by genotype and by agar concentration, and a factorial analysis of mixed data (FAMD) was performed for each group through the *FAMD* function in the R package *FactoMineR* [30]. This multivariate analysis allows us to analyse numerical and categorical variables and generates a multidimensional space where the first and second axes explain the largest proportion of variances as a combination of phenotypic traits (figure 2*b*,*c*; electronic supplementary material, table S2).

To characterize the trait-specific phenotypic variation, reaction norms were constructed connecting median values for each phenotypic trait. A linear regression fit was superimposed to visualize tendencies, although *p*-values were not significant in all cases (figure 3*a*; electronic supplementary material, figure S2 and table S3). A second set of reaction norms based on the coordinates of the first and second axes of the FAMD analyses as a statistical synthesis of phenotypic trait variation was also performed (figure 3*b*).

Finally, to test if the motility was altered in response to agar concentration or in response to genotype, and whether such variability might contribute to the observed developmental plasticity, we assessed colony growth over nutrient-rich substrates measuring the increase in its diameter and plotted in function of agar concentration (figure 4*a*) and of time (figure 4*b*). These values were normalized according to the colony diameter at 0 h. Then, we plotted the motility rate considering the proportion in which the colony diameter increases after 96 h and performed a two-way ANOVA to test the statistical significance of genotype and substrate conditions in such rate (figure 4*c*). Note that FBs do not develop in the presence of nutrients.

All analyses were conducted in R program (v. 3.2.3) through RStudio [31,32]. ggplot2 package v. 3.0.0 was employed for visualization [33].

## 3. Results

We observed that most of the phenotypic variation revealed in micrographs of figure 2*a* is explained by genotype (adj $R^2 = 0.571$), substrate stiffness (adj $R^2 = 0.166$) and their interaction (adj $R^2 = 0.185$). Statistical significance for each factor is strongly supported by the PERMANOVA analysis: pGenotype $= 9.9 \times 10^{-4}$, pEnvironment $= 9.9 \times 10^{-4}$, pInteraction $= 9.9 \times 10^{-4}$ (table 1). The FAMD analyses provided clarity in discriminating phenotypes. Specifically, by fixing each substrate condition (agar %) and assessing the differences among the genotypes, phenotypic differences can be distinguished (figure 2*b*). For example, all the genotypes (ellipses) are well distinguished at the lowest

**Figure 2.** Phenotypic variation of *M. xanthus*. (*a*) Phenotypes (population scale) micrographs with mature FBs (dark spots). Columns correspond with the genotypes (parental strain and knock-out mutants), rows correspond with different agar concentrations. FAMD multivariate analyses of phenotypic variation for (*b*) all genotypes per agar concentration, and (*c*) all agar concentrations per genotype. DZF1: parental strain; Δ*mkapC*, Δ*mkapA*/Δ*mkapC*, Δ*mkapA*, Δ*pktC2*/Δ*pktD1*: mutants; 0.5%, 1.0%, 1.5%, 2.0%, 2.5%: agar concentrations. 95% confidence interval ellipses enclose data centroids.

agar concentration (0.5%), but tend to overlap as agar concentration increases. However, the parental strain ellipse (DZF1) is clearly apart from that of the other phenotypic variants, except for 0.5%. Looking across agar concentrations, the arrangement of the genotypes follows equal directions along the *X*-axis (axis of major variation) in all agar concentrations, except for 0.5%, with larger contribution from the number of FBs and the standard distance between them (electronic supplementary material, table S2 and figure S3). Finally, while the Δ*mkapA/C* and Δ*mkapC* phenotypes overlap in all the cases, correlating with their similar phenotypic profiles shown in the micrographs, there is a non-additive effect of the double mutant, as it does not resemble the Δ*mkapA* phenotype.

By contrast, we observed more overlap when assessing agar concentrations (ellipses) for each genotype (figure 2*c*). There is not a clear sequence in the ellipse disposition and overlapping is present between almost all of the ellipses, except for 0.5% ellipse, which is well distinguished in all genotypes. We also did not observe a pattern of traits contributing to *X*- and *Y*-axes within each genotype, although ring formation is predominant in all cases (electronic supplementary material, table S2 and figure S3). Altogether, our results show that at 0.5% agar concentration, there are clear phenotypic differences within and among genotypes, and that phenotypic identity is blurred as agar concentration increases. Furthermore, the consistency of variables explaining the phenotypic variation at *X*-axis (Dim. 1) and the constancy in the sequence of genotype ellipses when considering each agar concentration suggest that the expressions of the genotype are enabled in a given environmental range. These patterns are notably robust across replicates, which is evident from the marginal

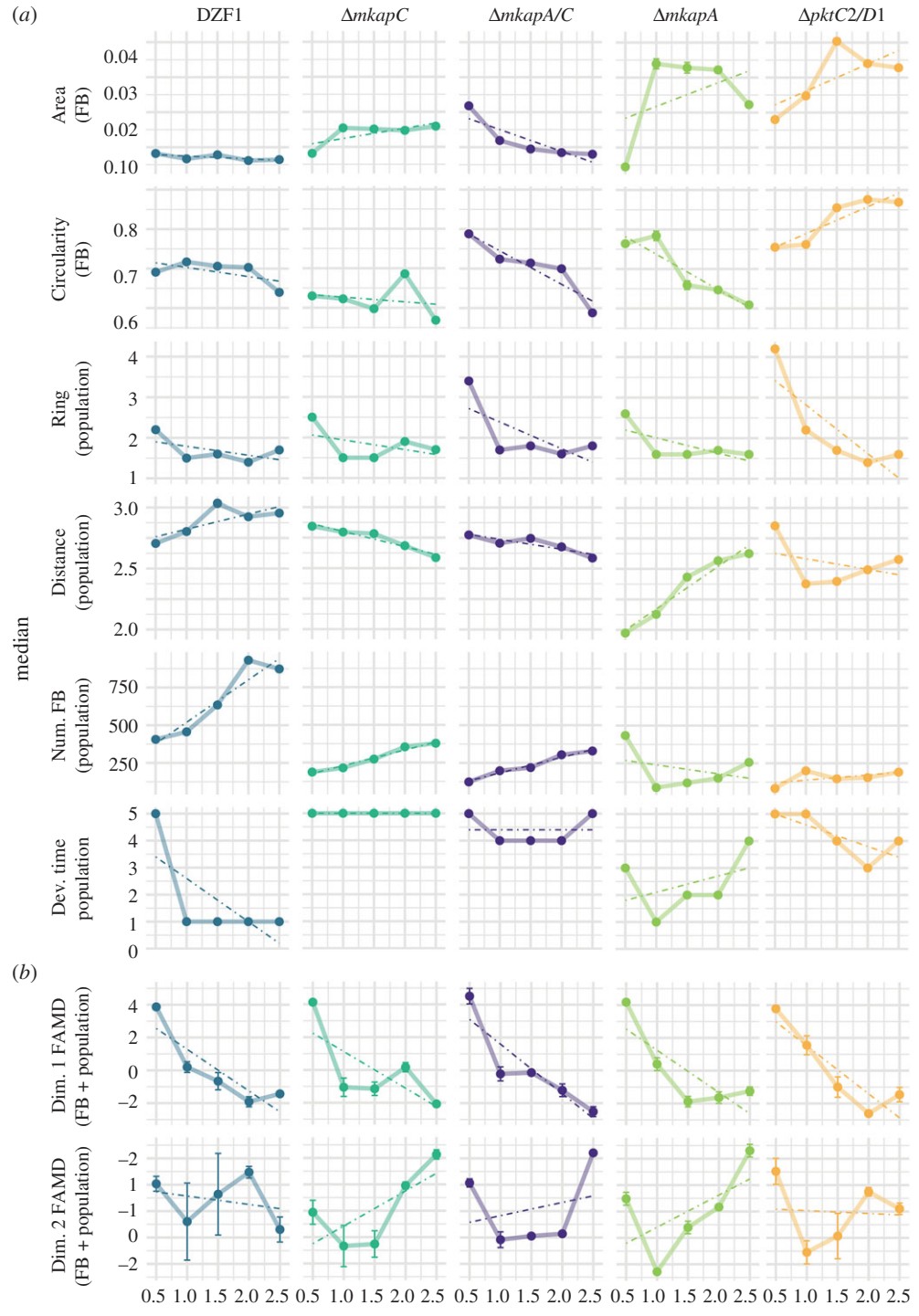

**Figure 3.** Phenotypic plasticity among *M. xanthus* parental and mutant genotypes in response to substrate agar concentration. Each line represents the reaction norm of a single genotype (columns) based on the median ± s.e. of (*a*) single phenotypic traits (rows) and (*b*) coordinates of dimension 1 and dimension 2 of the FAMD multivariate analysis. Linear regression fit with a dotted line, although *p*-values are not significant in all cases.

contribution of replicates to the overall phenotypic variation (electronic supplementary material, table S2 and figure S3).

Reaction norms show that there is substantial phenotypic variation among genotypes for individual traits across agar concentrations (figure 3*a*). Each trait×genotype combination has its own pattern, for both the single FB and population scales. For example, in Δ*mkapA*, 'Circularity' and 'Distance' are almost linear but with opposed slope signs, while 'Area' does not show a linear behaviour.

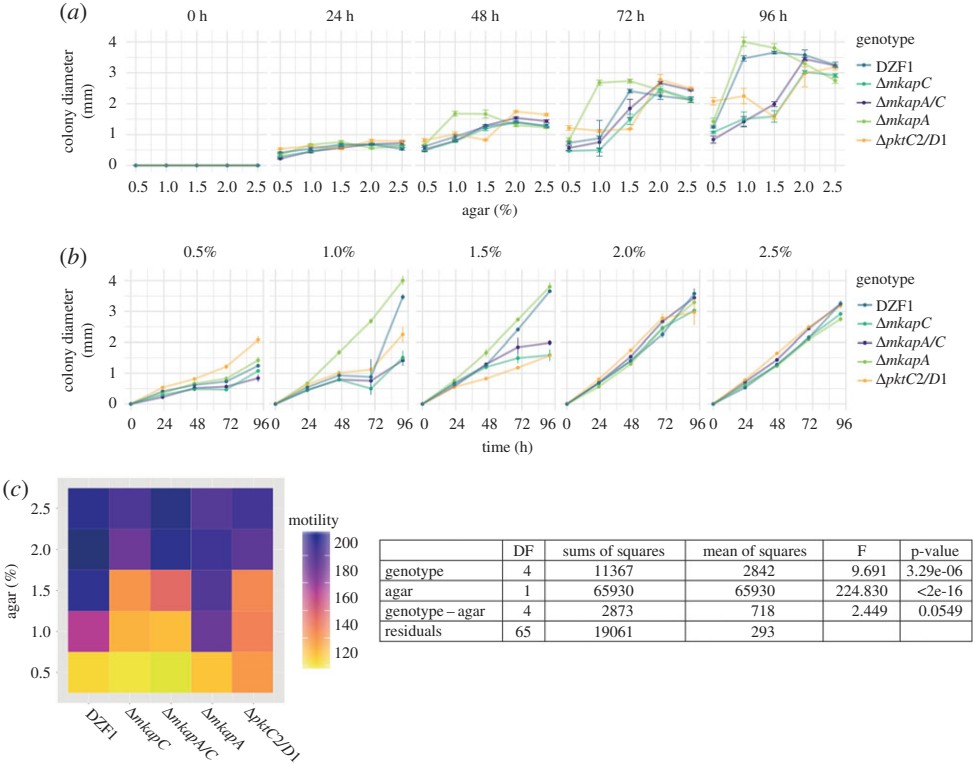

**Figure 4.** Colony growth rate of *M. xanthus* parental and mutant genotypes in response to agar concentration over nutrient-rich substrates. In the first two panels, dots connected by lines represent the median ± s.e. of the colony diameter (*a*) among time points from 0 to 96 h for each agar concentration, and (*b*) among agar concentration for each time point. Values were normalized according to colony diameter at 0 h. (*c*) Motility is approximated as the colony growth rate, considered as the proportion in which the colony diameter increases after 96 h, and depicted in the heatmap; darker colours represent high motility, while lighter ones represent low motility.

Remarkably, threshold variation is observed for some traits with particular condition and strain combinations, such as 'Ring' at the 0.5% condition. These effects are not general, especially for the parental strain (DZF1), in which the agar concentration is less strong in affecting 'Ring' and has no effect in FB traits and 'Dev. time', except at 0.5% agar. Finally, the large effect of substrate conditions, especially at the extreme ones (0.5% and 2.5%), is evident when plotting the FAMD coordinates as a statistical synthesis of phenotypic trait variation (figure 3*b*).

Regarding the colony growth rate in nutrient-rich substrates, there are strong interaction effects of agar concentration. At the lowest and highest agar concentrations, there was almost no difference among genotypes for colony growth rate (figure 4*b*,*c*). We observed a much lower motility on 0.5% compared to the rest of the agar concentrations (figure 4*b*,*c*), in line with previous reports [34]. Overall, when we track the colony growth, we observe that the strains slightly diverge according to their growth and that, as development proceeds, strains form two groups (DZF1 and Δ*mkapA* form one group and the rest of the strains another one (figure 4*a*,*c*)). However, we found that in such nutrient-rich substrates, the agar concentration is the most significant factor variable for the colony growth rate (see ANOVA results in figure 4*c*).

# 4. Discussion

We experimentally measured reaction norms of bacterial multicellular development across substrates with increasing stiffness. Phenotypic plasticity in *M. xanthus* development was revealed, involving changes in cell-to-substrate interaction due to increasing agar concentration in culture media plates. In this study, we focus on the potential for changes in morphology during development. These include changes in development time, spore maturation and FB size and number, which are important fitness-associated traits upon which selection may subsequently act. Previous studies with the same regulatory network reported statistically significant phenotypic differences when comparing mutants

with parental strain (DZF1), including differences in paradigmatic functional traits, such as spore count and viability [17]. Our current results, and those from previous studies, suggest that the joint effect of the physical environment and genetic background on morphologic and developmental traits is likely to be of evolutionary relevance.

The experimental design allowed for statistical discrimination of phenotypic changes at two scales: 'FB scale' and 'population scale'. In general, for each combination of genotype and substrate condition, FBs conforming the population exhibit little variation in their shape and size (figure 2*a*). Nevertheless, we found genotypic and substrate effects on phenotype at the 'population scale', which have been overlooked or dismissed when considering variation only at 'FB scale'. Remarkably, phenotypic identity over the genotypic and substrate conditions is not observed when considering only the FB scale (electronic supplementary material, figure S1; versus figure 2*b,c*).

To our knowledge, mechanisms behind the organization at the FB population scale have not been explored, although it is known that mechanical forces and living matter interact in a bidirectional way giving rise to robust patterns [13,14]. Based on our results and taking into consideration the influence of mechanisms such as substrate fibres alignment in colony formation [18] and the surface-tension-driven coarsening process of aggregation [35], we can suggest that the mechanical properties of the medium constitute a key element when thinking about environment at microscales. Also, substrate changes could determine the individual and collective motility, which could in turn drive aggregation. In this case, the developmental process occurring in nutrient-poor substrates would be mainly determined by the agar concentration. We found that the growth rate is largely determined by the agar concentration, with slight differences among strains (figure 4*a,b*). This is especially clear for the strain-independent low and high growth rate at the 0.5% and 2.5% conditions, respectively. Nevertheless, overall, the multicellular phenotypes at the FB and population scales are largely explained by agar concentration, genotype and the genotype–substrate interaction (table 1), which reflects the complexity of the developmental process under study.

Our results also demonstrate the value of investigating phenotypes from multi-level perspectives [11]. Phenotypic variation existed at both the level of single multicellular structures and the collective level of groups of these structures. It is important to note that specific environmental conditions have trait- and scale-specific effects. For instance, the substrate seems to play a much stronger role in certain ranges of agar concentration and for particular traits. We report a strain-independent ring formation, at population phenotype, surrounding the initial drop area at 0.5%. This could be explained by a higher cellular density at the edge of the drop forming when it dries at time 0 h, as in the coffee-ring formation physical phenomenon [36]. It is possible that at this agar concentration, cells cannot glide inwards easily, developing a ring of FBs at the edge. This hypothesis, and the precise mechanisms behind different phenotypes, remain to be tested.

In this work, we considered several points within the widest possible range of substrate modification which enabled a good approximation to the shape of reaction norms. We initially included a broader range of agar concentrations (electronic supplementary material, figure S4), but excluded those values where there was no development of FBs. This allowed us to notice that phenotypic change is usually greater at the extremes of the agar concentration range (figure 3*b*; in good agreement with previous theoretical proposals [37]). However, some limitations to this work and to the current study of plasticity in general should be considered. For instance, we modified stiffness by varying agar concentration, but other variables affecting the substrate properties such as water availability could be correlated. Indeed, reaction norms have provided useful perspectives regarding the paired relationship between a trait and an environmental factor for a specific moment in the developmental trajectory. However, new approaches considering the dynamical interaction among numerous phenotypic and environmental variables, across time and in ecologically meaningful ranges, are required. Moreover, plasticity is often studied in laboratory-adapted strains, and at least in this work, the parental laboratory-strain is not representative of the plastic responses for the rest of the genotypes (figures 1 and 2). Completely unexpected phenotypes may arise when considering natural environments or non-domesticated strains.

Overall, our study highlights the importance of the physical environment in the development of robust, yet plastic, aggregative microbial structures in different genotypic contexts. This in turn calls for systematic approaches to integrate the different factors and mechanisms (genetic and environmental) behind phenotypic plasticity and its consequent role in the emergence and evolution of multicellularity.

Data accessibility. Data are available from the Dryad Digital Repository: http://dx.doi.org/10.5061/dryad.308hs50 [28].
Authors' contributions. N.R.-Y., A.E.E., M.T. and M.B. conceived the study, N.R.-Y. and A.V.A. performed the experiments, all authors analysed the data and wrote the article.

Competing interests. We have no competing interests.

Funding. This study is funded by CONACYT (221341), DGAPA-PAPIIT-UNAM (RA105518) and PAPIIT-UNAM (IN111919).

Acknowledgements. N.R.-Y. is a doctoral student from Programa de Doctorado en Ciencias Biomédicas, Universidad Nacional Autónoma de México (UNAM) and received fellowship 580236 from CONACYT. Authors thank Marcelo Navarro-Díaz, Karen Carrasco-Espinosa, Alejandra Hernández-Terán, José Antonio Olivares Segura, Jorge Hernández-Cobos, Emilio Mora Van Cauwelaert and members of LANCIS for their support and valuable feedback. We also thank the International Centre for Theoretical Sciences (ICTS) for its support during the program Living Matter (Code: ICTS/Prog-LivingMatter2018/04), at the ICTS. Authors thank two anonymous reviewers for valuable comments and suggestions.

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
