## [Reviewer comments · Royal Society Open Science]

Plastic multicellular development of *Myxococcus xanthus*: genotype–environment interactions in a physical gradient

Natsuko Rivera-Yoshida, Alejandro V. Arzola, Juan A. Arias Del Angel, Alessio Franci,
Michael Travisano, Ana E. Escalante and Mariana Benítez

Article citation details

R. Soc. open sci. **6**: 181730.

<http://dx.doi.org/10.1098/rsos.181730>

Review timeline

Original submission: 16 October 2018

Revised submission: 14 February 2019

Final acceptance: 25 February 2019

Note: Reports are unedited and appear as
submitted by the referee. The review history
appears in chronological order.

Review History

RSOS-181730.R0 (Original submission)

Review form: Reviewer 1

Is the manuscript scientifically sound in its present form?

Yes

Are the interpretations and conclusions justified by the results?

Yes

Is the language acceptable?

Yes

Is it clear how to access all supporting data?

Yes

Do you have any ethical concerns with this paper?

No

Have you any concerns about statistical analyses in this paper?

No

Recommendation?

Major revision is needed (please make suggestions in comments)

Comments to the Author(s)

Review of RSOS-181730

Plastic multicellular development of *Myxococcus xanthus*: genotype-environment interactions in a physical gradient

This manuscript focuses on how abiotic conditions affect the multicellular development phenotypes of the social bacteria *M. xanthus*. Moreover, the authors test how development across the range of agar stiffnesses vary across genotypes, and statistically test for interactions between the abiotic and genetic treatments in a multitude of fruiting body morphologies. In general, the manuscript was easy to follow and clearly presented. The figures do a good job of showing the results of the experiment. It seems that the experimental procedures and the statistical analyses were done correctly. However, I outline some broad concerns below.

What are these *M. xanthus* mutants? There needs to be a more full explanation of why these mutants in particular were tested, and if there were any expectations with regard to the specific mutations and development across the abiotic gradient.

The authors also do not offer clear explanations or interpretations for the observed patterns. For example, is there any explanation for why the genotypes become less clustered on more soft agar? Similarly, any reason why you would expect agar stiffness to not have a major effect within a given genotype?

Is there any information or broader significance to the fruiting body traits that you measured? Many population level studies of *Myxococcus xanthus* focus on the fitness of strains by measuring spore production (e.g., work of the Velicer lab). Here, the authors do not measure any components of fitness (ie spore production). Do the authors expect that the phenotypic differences shown here correspond to any differences in spore production? If not, what is the importance of these results?

Overall, I think that this study was performed and presented well, and that there may be some interesting results. However, as it is currently written, interpreting what is interesting about the results is not clear. I believe that this could be published in Royal Society Open Science, but first I recommend that the authors revise the manuscript for resubmission focusing more on framing the study to make it more clear (1) why the particular mutants were used for the experiment, (2) the expectations/hypotheses for the experiment (3) the importance/significance of *M. xanthus* fruiting body morphology, and (4) some explanation of the broader relevance of this paper.

Review form: Reviewer 2**Is the manuscript scientifically sound in its present form?**

Yes

Are the interpretations and conclusions justified by the results?

Yes

Is the language acceptable?

Yes

Is it clear how to access all supporting data?

Yes

Do you have any ethical concerns with this paper?

No

Have you any concerns about statistical analyses in this paper?

No

Recommendation?

Major revision is needed (please make suggestions in comments)

Comments to the Author(s)

Plastic multicellular development of *Myxococcus xanthus*: genotype-environment interactions in a physical gradient

Summary:

The work carried out by Rivera-Yoshida et al. points to a fascinating question in evolutionary developmental biology: how different physical environments may affect cellular aggregation in the formation of multicellular structures, and whether the phenotypic variation of these structures that is caused by the changing environment have a similar magnitude and trajectory when different genotypic backgrounds are compared.

To address this question, the authors used the soil bacterium *M. xanthus*. They established and defined the changes in the environment by varying the agar concentration in the media on which the cells were plated to form starvation-induced multicellular fruiting bodies. That different substrates can influence fruiting body development was already known (e.g. (Escalante AE, et al. 2012, PLoS One), but Rivera-Yoshida et al. expand on previous work to study morphology in *M. xanthus* and designed an experiment that analyses environmental differences with a more rigorous and quantitative approach. The central result that both genotype and physical substrate (and their interaction) contribute to variation in both individual fruiting-body and broader population-level developmental phenotypes is clearly demonstrated. We were pleased to review this paper and hope that the authors find our comments helpful.

Major points:

The variation in genotypes is examined with one control strain DZF1 and four different mutants of DZF1 that are defective in the same signaling pathway known to affect fruiting body development. However, the mutants' biology is never discussed, particularly in light of how each mutation alters phenotypic plasticity. The paper therefore misses the opportunity to speculate regarding possible behavioral mechanisms underlying variable plasticity. If the authors consider such speculation to be insufficiently informed by what is known about the mutants, this could be explicitly noted.

The authors note that the paper's main result is partially expected from previous theoretical studies as well as from other papers that weren't specifically addressing the same question. For

example, differences in agar concentration are known to influence swarming rate differently across genotypic backgrounds. Since swarming ability is fundamental for proper fruiting body development, it is of interest for the authors to address whether the mutants show differences in motility and whether such variability might contribute to the observed plasticity. As briefly mentioned previously, the primary strength of the work is the approach they use to study morphological properties and their variations in different experimental conditions. Although these methods have been introduced long ago to implement the study of phenotypes (e.g. reaction norms, morphospaces), they still have been poorly used and applied on bacterial developmental and morphological properties.

In general, we support the publication of the scientific content that the reviewed paper contains. However, we would strongly suggest the authors to address all the major points, if not all the points we have listed, before publication.

We were glad to be able to review the paper from Rivera-Yoshida et al. and hope that our contribution with this revision will help the publishing processes.

Major points:

On the images processing and quantification:

For the following image reported in Fig 2a:

DZF1 and Δ mkapC - 0.5% seems to have more fruiting bodies that are not included in the image's frame. Are those fruiting bodies all included in the analysis or are they cropped out? Are these the fruiting bodies outside the initial spot?

For the following images reported in Fig 2a:

- Δ mkapC - from 1.0% to 2.5%
- Δ mkapA/C - 2.0% and 2.5%
- Δ pktC2/D1 - from 1.0% to 2.5%

All these grey-scale images show white regions mostly evident on the larger fruiting bodies, especially the larger ones, which. These white spots seem to may be caused by the light reflection from the illumination system. The presence of these spots could have a strongly bias on the quantification of geometrical parameters (e.g. area, circularity) that could in turn affect the following multivariate analyses as well. To control for the data accuracy, we would suggest the authors to show area quantifications that were done using the automated system and manual quantification of a subset of fruiting bodies that show such white features. In case of significant discrepancy, the images should be taken again with an illumination system that comes from below the plate and provides a homogeneous light throughout the entire sample.

Line 75 - Strains that are used in the manuscript - Throughout the entire manuscript, there is not a single reference to the physiological differences of that the selected mutants strains have if compared relative to the reference DZF1 parental strain. We strongly suggest that the authors describe to make clear what are the mutants they selected and why they were selected them. Nayar VT. and Inouye S. have shown strong effects of these mutants on fruiting body development and sporulation compared to the reference strain DZF1. Given such associations, a better contextualization of the mutants and their biological relevance to the paper is desired.

Line 86 - Here the authors refer to a FB as "fully developed" when it completes the process of darkening. Are there any known quantifiable correlates to darkening (e.g. spore counts, size) or precedents of this definition that could be cited? If this definition is unique to this manuscript,

this could be noted. This definition needs to be followed by a reference or a better explanation of what a fully developed FB is (spore counts? Size?). However, in case the authors are only presenting their definition of what a fully developed FB is (which is totally legit), I would suggest to be more direct and clear.

Line 96 - (FB out) it is hard to understand how the quantification was done here given the difficulties of tracking the spot margin after 96h. Was the spot margin quantified right after the culture dried at 0 h? How? The spot size can be very different after 96h on TPM plates.

Line 97 - (Maturation) see comment for Line 86. Also, moreover, in table S1, "Maturation" is defined as a binary population trait (0 and 1). Therefore, would one expect to see "Maturation" and "Scars" to show symmetric but opposite patterns to one another, as seems to be the case for only DZF1? If not, why not? And the definition of "Scars" could be clearer - (transparent, immature fruiting bodies?) - clear and not mature fruiting bodies(?) - to be symmetrical to one another, which is true only for DZF1 (Fig S2).

Line 97 - (Dev. time) Given the explanation in table S1, "Dev. Time" and "Maturation" seem to overlap. What distinguishes them is not clear from the definition that has been provided.

Line 128 - Given the molecular background, isn't the overlap of observation that Δ mkpC and Δ mkpA/C overlap is as important as the one which would point to just as significant as the absence of overlap between Δ mkpA/C and Δ mkpA in low agar concentration (0.5% to 1.5%)? Why the emphasis on the latter?. However, this aspect raises the question on the biological meaning of such similarities/dissimilarities that could be discussed.

Line 130 - The location of the DZF1 ellipse is interesting and could be discussed more. The relative position of DZF1 position seems to change drastically from 0.5% to any the other agar concentrations. The position that the DZF1 ellipse occupies at higher agar concentrations doesn't vary as dramatically as the authors seem to suggest reported. However, it is striking that a similar trend, yet almost symmetrical to the one of DZF1, is shown by Δ pktC2/D1. As for the previous comment, the biological relevance of this trend could be addressed expanded in the discussion.

Line 148 - The authors mention "Shape" but it is not clear to what parameter/s they are referring to in Figure 3. If they are referring to "Circularity", it would help they should be consistent with the nomenclature present in the figure. Also, we suggest using 'linear' in the same sentence, the author use the adjective 'lineal' instead of linear? If so, linear would be a more appropriate term in this case. Thus, when the author use the adjective linear in the next sentence when commenting on the trend of the trait "Area" the comparison will result to be clearer. for consistency with the next sentence.

One very surprising result piece of data that I have found particularly surprising is the reported relationship between number of fruiting bodies in the light of the variation of the "Area" and the "area" parameter. Interestingly, DZF1 shows an almost steady increase in growth in the number of FBs as if agar concentration increases. Yet, especially when compared to the mutant strains, the area is not increasing nor decreasing without any corresponding change in the average area of individual FBs. This observation seems quite unexpected. Given the same number of cells (cell density is kept consistent in all experimental conditions), DZF1 forms up to 1000 FBs and yet average the FB area is barely decreased if not at all. Moreover, the measurement "Distance" parameter is also behaving unexpectedly. One expects that with an increase in FBs number and with relatively constant "Area" Area should cause that doesn't show differences, the variable "Distance" variable should decrease - or at least not increase at all -- yet this does not appear to be the case as the current plot doesn't show (for DZF1). We think in case these

quantifications will should be rechecked be confirmed by the authors. If they are confirmed as accurate,, they should be discussed with possible explanations due to their surprising character, as they developed further within the paper discussion. In fact, such observations canmay be of interest to the study of the development of multicellular structures in a changing environments.

Additional points:

Line 36-38 - The reference from Sultas S. 2015 (ref #1 in the paper), could go at the end of the sentence given its relevance to all expressed concepts. I wouldWe suggest to adding more experimental paper references to the first part of the introduction (e.g. Monteiro et al 2013, Matsumoto et al 2013, Mayakawa et al 2010, Gilbert SF 2016) and not limit the material references source almost to reviews only.

Here few suggestions: - Monteiro et al 2013, Matsumoto et al 2013, Mayakawa et al 2010, Gilbert SF 2016,

Line 44 - I would stronglyWe suggest recommend the authors to cite West-Eberhard MJ's book "Developmental Plasticity and Evolution. 2005" since this book (and other publications from the same author during the same years) brought to debate and research more extensively the concept the authors express in this paragraph.

Line 46 - the authors write "The origin of multicellularity is a major evolutionary transition." I would be more specific adding that it is a "[...] major evolutionary transition in organismal development".

Line 52 - "Importantly, physical processes and mechanical forces associated...". I would suggest toPlease rephrase this sentence. Right now, it is not clear whyto clarify what distinction between mechanical forces are distinguished and not consideredand physical processes is intended (or are they?).

Line 93 - The authors refer to ImageJ v2.0.0 in the text butand cite FIJI (ref.# 16 in the manuscript).

Line 108 - It seems would find more appropriate to refer to Sup. Table 2 and notrather than Sup. Table 1. Scree plots and a plot showing the contribution per dimensions would greatly help the reader in accessing the information present in Sup. Table 2.

Line 113 - Do all the measured traits have a symmetric distribution around the mean? If some are skewed, it maywould be better to use medians instead.

Figures - Color codes in figures could be chosen to be more "color-blind friendly". Specifically, Fig. 2b and Fig. 2c are very hard to navigate if the reader is affected by deuteranopia (same for Sup Fig 1). The authors may find useful the software Color Oracle (colororacle.org) to help testing the images for color- blind readers. In addition, since the author are using ggplot2 package in R to generate images, they can make use of the `colorblind_pal()` function:
<https://rdrr.io/cran/ggthemes/man/colorblind.html>

Line 183 - It would be interesting to visualize such data in a representative panel (no quantifications needed).

Fig 3 - We suggest reportingIn this case, an inferential error (e.g. standard error or CI) should be preferred torather than the standard deviationerror (descriptive error).

Fig S1 - Legend text explaining graph features, e.g. the meaning of colors, is missing.

Fig S2 – No explanation of missing error bars. exp.

Table S1 – “Circularity” should be noted as dimensionless.

Decision letter (RSOS-181730.R0)

14-Jan-2019

Dear Ms Rivera-Yoshida,

The editors assigned to your paper ("Plastic multicellular development of *Myxococcus xanthus*: genotype-environment interactions in a physical gradient") have now received comments from reviewers. We would like you to revise your paper in accordance with the referee and Associate Editor suggestions which can be found below (not including confidential reports to the Editor). Please note this decision does not guarantee eventual acceptance.

Please submit a copy of your revised paper before 06-Feb-2019. Please note that the revision deadline will expire at 00.00am on this date. If we do not hear from you within this time then it will be assumed that the paper has been withdrawn. In exceptional circumstances, extensions may be possible if agreed with the Editorial Office in advance. We do not allow multiple rounds of revision so we urge you to make every effort to fully address all of the comments at this stage. If deemed necessary by the Editors, your manuscript will be sent back to one or more of the original reviewers for assessment. If the original reviewers are not available, we may invite new reviewers.

- Data accessibility

It is a condition of publication that all supporting data are made available either as supplementary information or preferably in a suitable permanent repository. The data

accessibility section should state where the article's supporting data can be accessed. This section should also include details, where possible of where to access other relevant research materials such as statistical tools, protocols, software etc can be accessed. If the data have been deposited in an external repository this section should list the database, accession number and link to the DOI for all data from the article that have been made publicly available. Data sets that have been deposited in an external repository and have a DOI should also be appropriately cited in the manuscript and included in the reference list.

If you wish to submit your supporting data or code to Dryad (<http://datadryad.org/>), or modify your current submission to dryad, please use the following link:
<http://datadryad.org/submit?journalID=RSOS&manu=RSOS-181730>

- **Competing interests**

- **Authors' contributions**

- **Acknowledgements**

- **Funding statement**

on behalf of Professor Joanne Santini (Associate Editor) and Professor Kevin Padian (Subject Editor)
openscience@royalsociety.org

Comments to Author:

Reviewers' Comments to Author:

Reviewer: 1

Comments to the Author(s)

Review of RSOS-181730

Plastic multicellular development of *Myxococcus xanthus*: genotype-environment interactions in a physical gradient

This manuscript focuses on how abiotic conditions affect the multicellular development phenotypes of the social bacteria *M. xanthus*. Moreover, the authors test how development across the range of agar stiffnesses vary across genotypes, and statistically test for interactions between the abiotic and genetic treatments in a multitude of fruiting body morphologies. In general, the manuscript was easy to follow and clearly presented. The figures do a good job of showing the results of the experiment. It seems that the experimental procedures and the statistical analyses were done correctly. However, I outline some broad concerns below.

What are these *M. xanthus* mutants? There needs to be a more full explanation of why these mutants in particular were tested, and if there were any expectations with regard to the specific mutations and development across the abiotic gradient.

The authors also do not offer clear explanations or interpretations for the observed patterns. For example, is there any explanation for why the genotypes become less clustered on more soft agar? Similarly, any reason why you would expect agar stiffness to not have a major effect within a given genotype?

Is there any information or broader significance to the fruiting body traits that you measured? Many population level studies of *Myxococcus xanthus* focus on the fitness of strains by measuring spore production (e.g., work of the Velicer lab). Here, the authors do not measure any components of fitness (ie spore production). Do the authors expect that the phenotypic differences shown here correspond to any differences in spore production? If not, what is the importance of these results?

Overall, I think that this study was performed and presented well, and that there may be some interesting results. However, as it is currently written, interpreting what is interesting about the results is not clear. I believe that this could be published in Royal Society Open Science, but first I recommend that the authors revise the manuscript for resubmission focusing more on framing the study to make it more clear (1) why the particular mutants were used for the experiment, (2) the expectations/hypotheses for the experiment (3) the importance/significance of *M. xanthus* fruiting body morphology, and (4) some explanation of the broader relevance of this paper.

Reviewer: 2

Comments to the Author(s)

Plastic multicellular development of *Myxococcus xanthus*: genotype-environment interactions in a physical gradient

Summary:

The work carried out by Rivera-Yoshida et al. points to a fascinating question in evolutionary

developmental biology: how different physical environments may affect cellular aggregation in the formation of multicellular structures, and whether the phenotypic variation of these structures that is caused by the changing environment have a similar magnitude and trajectory when different genotypic backgrounds are compared.

To address this question, the authors used the soil bacterium *M. xanthus*. They established and defined the changes in the environment by varying the agar concentration in the media on which the cells were plated to form starvation-induced multicellular fruiting bodies. That different substrates can influence fruiting body development was already known (e.g. (Escalante AE, et al. 2012, PLoS One), but Rivera-Yoshida et al. expand on previous work to study morphology in *M. xanthus* and designed an experiment that analyses environmental differences with a more rigorous and quantitative approach. The central result that both genotype and physical substrate (and their interaction) contribute to variation in both individual fruiting-body and broader population-level developmental phenotypes is clearly demonstrated. We were pleased to review this paper and hope that the authors find our comments helpful.

Major points:

The variation in genotypes is examined with one control strain DZF1 and four different mutants of DZF1 that are defective in the same signaling pathway known to affect fruiting body development. However, the mutants' biology is never discussed, particularly in light of how each mutation alters phenotypic plasticity. The paper therefore misses the opportunity to speculate regarding possible behavioral mechanisms underlying variable plasticity. If the authors consider such speculation to be insufficiently informed by what is known about the mutants, this could be explicitly noted.

The authors note that the paper's main result is partially expected from previous theoretical studies as well as from other papers that weren't specifically addressing the same question. For example, differences in agar concentration are known to influence swarming rate differently across genotypic backgrounds. Since swarming ability is fundamental for proper fruiting body development, it is of interest for the authors to address whether the mutants show differences in motility and whether such variability might contribute to the observed plasticity. As briefly mentioned previously, the primary strength of the work is the approach they use to study morphological properties and their variations in different experimental conditions. Although these methods have been introduced long ago to implement the study of phenotypes (e.g. reaction norms, morphospaces), they still have been poorly used and applied on bacterial developmental and morphological properties.

In general, we support the publication of the scientific content that the reviewed paper contains. However, we would strongly suggest the authors to address all the major points, if not all the points we have listed, before publication.

We were glad to be able to review the paper from Rivera-Yoshida et al. and hope that our contribution with this revision will help the publishing processes.

Major points:

On the images processing and quantification:

For the following image reported in Fig 2a:

DZF1 and Δ mkapC - 0.5% seems to have more fruiting bodies that are not included in the image's frame. Are those fruiting bodies all included in the analysis or are they cropped out? Are these the fruiting bodies outside the initial spot?

For the following images reported in Fig 2a:

- mkapC - from 1.0% to 2.5%
- mkapA/C - 2.0% and 2.5%
- pktC2/D1 - from 1.0% to 2.5%

All these grey-scale images show white regions mostly evident on the larger fruiting bodies, especially the larger ones, which. These white spots seem to may be caused by the light reflection from the illumination system. The presence of these spots could have a strongly bias on the quantification of geometrical parameters (e.g. area, circularity) that could in turn affect the following multivariate analyses as well. To control for the data accuracy, we would suggest the authors to show area quantifications that were done using the automated system and manual quantification of a subset of fruiting bodies that show such white features. In case of significant discrepancy, the images should be taken again with an illumination system that comes from below the plate and provides a homogeneous light throughout the entire sample.

Line 75 - Strains that are used in the manuscript - Throughout the entire manuscript, there is not a single reference to the physiological differences of that the selected mutants strains have if compared relative to the reference DZF1 parental strain. We strongly suggest that the authors describe to make clear what are the mutants they selected and why they were selected them. Nayar VT. and Inouye S. have shown strong effects of these mutants on fruiting body development and sporulation compared to the reference strain DZF1. Given such associations, a better contextualization of the mutants and their biological relevance to the paper is desired.

Line 86 - Here the authors refer to a FB as “fully developed” when it completes the process of darkening. Are there any known quantifiable correlates to darkening (e.g. spore counts, size) or precedents of this definition that could be cited? If this definition is unique to this manuscript, this could be noted. This definition needs to be followed by a reference or a better explanation of what a fully developed FB is (spore counts? Size?). However, in case the authors are only presenting their definition of what a fully developed FB is (which is totally legit), I would suggest to be more direct and clear.

Line 96 - (FB out) it is hard to understand how the quantification was done here given the difficulties of tracking the spot margin after 96h. Was the spot margin quantified right after the culture dried at 0 h? How? The spot size can be very different after 96h on TPM plates.

Line 97 - (Maturation) see comment for Line 86. Also, Moreover, in table S1, “Maturation” is defined as a binary population trait (0 and 1). Therefore, I would expect to see “Maturation” and “Scars” to show symmetric but opposite patterns to one another, as seems to be the case for only DZF1? If not, why not? And the definition of “Scars” could be clearer - (transparent, immature fruiting bodies?) - clear and not mature fruiting bodies(?) - to be symmetrical to one another, which is true only for DZF1 (Fig S2).

Line 97 - (Dev. time) Given the explanation in table S1, “Dev. Time” and “Maturation” seem to overlap. What distinguishes them is not clear from the definition that has been provided.

Line 128 - Given the molecular background, isn't the overlap of observation that □mkapC and □mkapA/C overlap is as important as the one which would point to just as significant as the absence of overlap between □mkapA/C and □mkapA in low agar concentration (0.5% to 1.5%)? Why the emphasis on the latter? However, this aspect raises the question on the biological meaning of such similarities/dissimilarities that could be discussed.

Line 130 – The location of the DZF1 ellipse is interesting and could be discussed more. The relative position of DZF1 position seems to change drastically from 0.5% to any the other agar concentrations. The position that the DZF1 ellipse occupies at higher agar concentrations doesn't vary as dramatically as the authors seem to suggest reported. However, it is striking that a similar trend, yet almost symmetrical to the one of DZF1, is shown by $\square_{\text{pktC2/D1}}$. As for the previous comment, the biological relevance of this trend could be addressed expanded in the discussion.

Line 148 – The authors mentions “sShape” but it is not clear to what parameter/s they are referring to in Figure 3. If they are referring to “Circularity”, it would help they should be consistent with the nomenclature present in the figure. Also, we suggest using ‘linear’ in the same sentence, the author use the adjective ‘lineal’ instead of linear? If so, linear would be a more appropriate term in this case. Thus, when the author use the adjective linear in the next sentence when commenting on the trend of the trait “Area” the comparison will result to be clearer. for consistency with the next sentence.

One very surprising result piece of data that I have found particularly surprising is the reported relationship between number of fruiting bodies in the light of the variation of the “Area” and the “area” parameter. Interestingly, DZF1 shows an almost steady increase growth in the number of FBs as if agar concentration increases. Yet, especially when compared to the mutant strains, the area is not increasing nor decreasing without any corresponding change in the average area of individual FBs. This observation seems quite unexpected. Given the same number of cells (cell density is kept consistent in all experimental conditions), DZF1 forms up to 1000 FBs and yet average the FB area is barely decreased if not at all. Moreover, the measurement “Distance” parameter is also behaving unexpectedly. One expects that with an increase in FBs number and with relatively constant “Area” Area should cause that doesn't show differences, the variable “Distance” variable to should decrease - or at least not increase at all -- yet this does not appear to be the case as the current plot doesn't show (for DZF1). We think in case these quantifications will should be rechecked be confirmed by the authors. If they are confirmed as accurate,, they should be discussed with possible explanations due to their surprising character, as they developed further within the paper discussion. In fact, such observations can may be of interest to the study of the development of multicellular structures in a changing environments.

Additional points:

Line 36-38 - The reference from Sultas S. 2015 (ref #1 in the paper), could go at the end of the sentence given its relevance to all expressed concepts. I would suggest adding more experimental paper references to the first part of the introduction (e.g. Monteiro et al 2013, Matsumoto et al 2013, Mayakawa et al 2010, Gilbert SF 2016) and not limit the material references source almost to reviews only.

Here few suggestions: - Monteiro et al 2013, Matsumoto et al 2013, Mayakawa et al 2010, Gilbert SF 2016,

Line 44 – I would strongly suggest recommend the authors to cite West-Eberhard MJ's book “Developmental Plasticity and Evolution. 2005” since this book (and other publications from the same author during the same years) brought to debate and research more extensively the concept the authors express in this paragraph.

Line 46 – the authors write “The origin of multicellularity is a major evolutionary transition.” I would be more specific adding that it is a “[...] major evolutionary transition in organismal development”.

Line 52 – “Importantly, physical processes and mechanical forces associated...”. I would suggest to please rephrase this sentence. Right now, it is not clear why to clarify what distinction between mechanical forces are distinguished and not considered and physical processes is intended (or are they?).

Line 93 – The authors refer to ImageJ v2.0.0 in the text but and cite FIJI (ref.# 16 in the manuscript).

Line 108 – It seems would find more appropriate to refer to Sup. Table 2 and not rather than Sup. Table 1. Scree plots and a plot showing the contribution per dimensions would greatly help the reader in accessing the information present in Sup. Table 2.

Line 113 – Do all the measured traits have a symmetric distribution around the mean? If some are skewed, it may be better to use medians instead.

Figures - Color codes in figures could be chosen to be more “color-blind friendly”. Specifically, Fig. 2b and Fig. 2c are very hard to navigate if the reader is affected by deuteranopia (same for Sup Fig 1). The authors may find useful the software Color Oracle (colororacle.org) to help testing the images for color-blind readers. In addition, since the authors are using ggplot2 package in R to generate images, they can make use of the `colorblind_pal()` function:
<https://rdrr.io/cran/ggthemes/man/colorblind.html>

Line 183 – It would be interesting to visualize such data in a representative panel (no quantifications needed).

Fig 3 – We suggest reporting. In this case, an inferential error (e.g. standard error or CI) should be preferred to rather than the standard deviation error (descriptive error).

Fig S1 – Legend text explaining graph features, e.g. the meaning of colors, is missing.

Fig S2 – No explanation of missing error bars. exp.

Table S1 – “Circularity” should be noted as dimensionless.

Author's Response to Decision Letter for (RSOS-181730.R0)

See Appendix A.

Decision letter (RSOS-181730.R1)

25-Feb-2019

Dear Ms Rivera-Yoshida,

I am pleased to inform you that your manuscript entitled "Plastic multicellular development of *Myxococcus xanthus*: genotype-environment interactions in a physical gradient" is now accepted for publication in Royal Society Open Science.

on behalf of Professor Joanne Santini (Associate Editor) and Professor Kevin Padian (Subject Editor)
openscience@royalsociety.org

Follow Royal Society Publishing on Twitter: [@RSocPublishing](https://twitter.com/RSocPublishing)

Appendix A

Dear Professor Jeremy Sanders CBE FRS,

We appreciate the valuable and insightful comments made by the reviewers, which have significantly contributed to an improved version of the article. We have addressed all the comments below and worked on a revised version that includes the reviewer's suggestions.

We look forward to hearing from you and hope that this revised version is suitable for publication in Royal Society Open Science.

Best regards,

The authors

Reviewer: 1

Comments to the Author(s)

Review of RSOS-181730

Plastic multicellular development of *Myxococcus xanthus*: genotype-environment interactions in a physical gradient

This manuscript focuses on how abiotic conditions affect the multicellular development phenotypes of the social bacteria *M. xanthus*. Moreover, the authors test how development across the range of agar stiffnesses vary across genotypes, and statistically test for interactions between the abiotic and genetic treatments in a multitude of fruiting body morphologies. In general, the manuscript was easy to follow and clearly presented. The figures do a good job of showing the results of the experiment. It seems that the experimental procedures and the statistical analyses were done correctly. However, I outline some broad concerns below.

1. What are these *M. xanthus* mutants? There needs to be a more full explanation of why these mutants in particular were tested, and if there were any expectations with regard to the specific mutations and development across the abiotic gradient.

*This is indeed an important issue. DZF1 is one of the Myxococcus xanthus standard laboratory strains. $\Delta mkapC$, $\Delta mkapA/\Delta mkapC$, $\Delta mkapA$, $\Delta pktC2/\Delta pktD1$ are DZF1 derived in-frame deletion mutants of the network of PSTKs and the associated scaffold proteins Mkaps, which participate during the development of FBs (Arias Del Angel et al., 2017, 2018). An interesting feature of these mutants is that they do not arrest development and thus allow phenotypic changes to be tracked at the multicellular scale. Previous studies suggest that deletion of the *pktC2*, *pktD1* and other components of this network (*pktA2*, *pktD9*) impact development and fruiting body formation (Escalante et al., 2015; Nariya & Inouye, 2002, 2003, 2005, 2006). *MkapA* and *MkapC* are scaffold proteins in this network, but there are not previous reports regarding the phenotypic consequences of their deletion.*

This information has now been included in the description of the biological material in the Methods section.

2. The authors also do not offer clear explanations or interpretations for the observed patterns. For example, is there any explanation for why the genotypes become less clustered on more soft agar? Similarly, any reason why you would expect agar stiffness to not have a major effect within a given genotype?

Thank you for your insightful questions.

In order to further understand why genotypes behave differently on different substrates, we assessed the colony growth rate on nutrient-rich substrates by measuring the diameter of the colony in different time points (but please note that FBs do not develop in the presence of nutrients). We considered this rate as a proxy to motility and observed a much lower motility rate on 0.5% compared to the rest of the agar concentrations. However, our observations at the single-cell level suggest that motility is not arrested in low agar concentrations. This behaviour on soft agar is in line with previous reports (Shi & Zusman, 1992).

Overall, when we track the colony growth in nutrient-rich substrate we observe that the strains slightly differentiate and that, as development proceeds, strains form two groups (DZF1 and Δ mkapA form one group and the rest of the strains another one). However, we found that in such nutrient-rich substrates, it is the agar concentration that mostly drives the colony expansion rates (see new Figure 4 in the main text). Since the growth rate of the colony in nutrient-rich substrates somehow reflects the individual or collective motility in the specific strains, we could speculate that the motility in the developmental process occurring in nutrient-poor substrates is mainly affected by the agar concentration. Nevertheless, the multicellular phenotypes at the FB and population scales are certainly explained by a significant contribution the agar concentration, but also of by the genotype and the genotype-substrate interaction (Table 1 and Figure 4(c) in the main text).

Finally, it is important to note that specific environmental conditions have trait- and scale-specific effects. For instance, the substrate seems to play a much stronger role in certain ranges of agar concentration and for particular traits. We report a strain-independent ring formation at the edge of the colony at 0.5%. This could be explained by a higher cellular density at the edge of the drop forming when it dries at time 0 h, as in the coffee-ring formation physical phenomenon (Andac et al., 2019). It is possible that at this agar concentration, cells cannot glide inwards easily, developing a ring of FBs at the edge, where there is a higher cellular density, and leaving immature aggregates at the center of the drop. This hypothesis, and the precise mechanisms behind different phenotypes remain to be tested.

The genetic identity of the strains and the results regarding their motility have now been included in the Methods section, as a new figure (Figure 4), a new table (Table 2) and as whole new paragraph at the end of the Results section and in the Discussion.

Regarding your second question, the relative location of the DZF1 with respect to other genotypes is very interesting and has led us to speculate about the potential canalization or domestication of the strain. Indeed, other work has suggested that standard laboratory strains exhibit canalization (low plasticity) in important traits (Palková, 2004). This could be the case for DZF1, but at this point we have too little evidence to even suggest it. It would be necessary to compare it with wild strains -not mutants derived from the same strain- in order to further test this idea. In the current version we prefer to just say that "...plasticity is often studied in lab-adapted strains, and at least in this work, the parental lab-strain is not representative of the plastic responses for the rest of the genotypes (figures 1 and 2)."

3. Is there any information or broader significance to the fruiting body traits that you measured? Many population level studies of *Myxococcus xanthus* focus on the fitness of strains by measuring spore production (e.g., work of the Velicer lab). Here, the authors do not measure any components of fitness (i.e. spore production). Do the authors expect that the phenotypic differences shown here correspond to any differences in spore production? If not, what is the importance of these results?

Thank you for your insightful comment. In this study we focus on the potential for morphologic phenotypic changes, but we do observe changes ranging from modest to dramatic upon which selection may subsequently act. These include changes in development time, spore maturation and, more indirectly, FB size and number. Moreover, it has been reported that mutants of the same regulatory network exhibit statistically significant phenotypic differences when comparing mutants with parental strain (DZF1), including differences in spore count and viability (Escalante et al., 2012). This suggests that the joint effect of the physical environmental and the genetic background on morphologic and developmental traits may also be of evolutionary relevance. We now mention this in the Discussion section.

4. Overall, I think that this study was performed and presented well, and that there may be some interesting results. However, as it is currently written, interpreting what is interesting about the results is not clear. I believe that this could be published in Royal Society Open Science, but first I recommend that the authors revise the manuscript for resubmission focusing more on framing the study to make it more clear (1) why the particular mutants were used for the experiment, (2) the expectations/hypotheses for the experiment (3) the importance/significance of *M. xanthus* fruiting body morphology, and (4) some explanation of the broader relevance of this paper.

Thank you very much for your positive feedback. We have addressed all of these issues in the (1) methods section, (2) introduction (next-to-last paragraph), (3) discussion (first paragraph), and (4) along the revised text.

Reviewer: 2

The work carried out by Rivera-Yoshida et al. points to a fascinating question in evolutionary developmental biology: how different physical environments may affect cellular aggregation in the formation of multicellular structures, and whether the phenotypic variation of these structures that is caused by the changing environment have a similar magnitude and trajectory when different genotypic backgrounds are compared.

To address this question, the authors used the soil bacterium *M. xanthus*. They established and defined the changes in the environment by varying the agar concentration in the media on which the cells were plated to form starvation-induced multicellular fruiting bodies. That different substrates can influence fruiting body development was already known (e.g. (Escalante AE, et al. 2012, PLoS One), but Rivera-Yoshida et al. expand on previous work to study morphology in *M. xanthus* and designed an experiment that analyses environmental differences with a more rigorous and quantitative approach. The central result that both genotype and physical substrate (and their interaction) contribute to variation in both individual fruiting-body and broader population-level developmental phenotypes is clearly demonstrated. We were pleased to review this paper and hope that the authors find our comments helpful.

Major points:

1. The variation in genotypes is examined with one control strain DZF1 and four different mutants of DZF1 that are defective in the same signaling pathway known to affect fruiting body development. However, the mutants' biology is never discussed, particularly in light of how each mutation alters phenotypic plasticity. The paper therefore misses the opportunity to speculate regarding possible behavioral mechanisms underlying variable plasticity. If the authors consider such speculation to be insufficiently informed by what is known about the mutants, this could be explicitly noted.

*Thank you for this important observation. This is indeed an important issue. DZF1 is one of the Myxococcus xanthus standard laboratory strains. $\Delta mkapC$, $\Delta mkapA/\Delta mkapC$, $\Delta mkapA$, $\Delta pktC2/\Delta pktD1$ are DZF1 derived in-frame deletion mutants of the network of PSTKs and the associated scaffold proteins Mkaps, which participate during the development of FBs (Arias Del Angel et al., 2017, 2018). An interesting feature of these mutants is that they do not arrest development and thus allow to track phenotypic changes at the multicellular scale. Previous studies suggest that deletion of the *pktC2*, *pktD1* and other components of this network (*pktA2*, *pktD9*) impact development and fruiting body formation (Escalante et al., 2015; Nariya & Inouye, 2002, 2003, 2005, 2006). *MkapA* and *MkapC* are scaffold proteins in this network, but there are not previous reports regarding the phenotypic consequences of their deletion.*

This information has now been included in the description of the biological material in the Methods section.

2. The authors note that the paper's main result is partially expected from previous theoretical studies as well as from other papers that weren't specifically addressing the same question. For example, differences in agar concentration are known to influence swarming rate differently across genotypic backgrounds. Since swarming ability is fundamental for proper fruiting body development, it is of interest for the authors to address whether the mutants show differences in motility and whether such variability in might contribute to the observed plasticity . As briefly mentioned previously, the primary

strength of the work is the approach they use to study morphological properties and their variations in different experimental conditions. Although these methods have been introduced long ago to implement the study of phenotypes (e.g. reaction norms, morphospaces), they still have been poorly used and applied on bacterial developmental and morphological properties.

This is a very interesting point. In order to address this question, we assessed the colony growth rate on nutrient-rich substrates by measuring the diameter of the colony in different time points (but please note that FBs do not develop in the presence of nutrients). We observed a much lower motility on 0.5% compared to the rest of the agar concentrations. This is in line with previous reports (Shi & Zusman, 1992).

Overall, when we track the colony growth in nutrient-rich substrate we observe that the strains slightly differentiate according to their expansion rate. As development proceeds, strains form two groups (DZF1 and Δ mkapA form one group and the rest of the strains another one). However, we found that in such nutrient-rich substrates, it is the agar concentration that mostly drives the colony expansion rates (see new Figure 4 in the main text). Since the expansion rate of the colony in nutrient-rich substrates somehow reflects the individual or collective motility in the specific strains, we could speculate that the motility in the developmental process occurring in nutrient-poor substrates is mainly affected by the agar concentration. Nevertheless, the multicellular phenotypes at the FB and population scales are certainly explained by a significant contribution the agar concentration, but also of by the genotype and the genotype-substrate interaction (Table 1 and Figure 4 in the main text).

Finally, it is important to note that specific environmental conditions have trait- and scale-specific effects. For instance, the substrate seems to play a much stronger role in certain ranges of agar concentration and for particular traits. We report a strain-independent ring formation at the edge of the colony at 0.5%. This could be explained by a higher cellular density at the edge of the drop forming when it dries at time 0 h, as in the coffee-ring formation physical phenomenon (Andac et al., 2019). It is possible that at this agar concentration, cells cannot glide inwards easily, developing a ring of FBs at the edge, where there is a higher cellular density, and leaving immature aggregates at the center of the drop. This hypothesis, and the precise mechanisms behind different phenotypes remain to be tested.

The identity of the strains and these results have now been included in the Methods section, as a new figure (Figure 4) and as whole new paragraph at the end of the Results section and in the Discussion.

In general, we support the publication of the scientific content that the reviewed paper contains. However, we would strongly suggest the authors to address all the major points, if not all the points we have listed, before publication. We were glad to be able to review the paper from Rivera-Yoshida et al. and hope that our contribution with this revision will help the publishing processes.

We truly appreciate your insightful and thorough review, which has certainly helped the publishing and research process.

Major points:

On the images processing and quantification:

3. For the following image reported in Fig 2a:

DZF1 and mkapC – 0.5% seems to have more fruiting bodies that are not included in the image's frame. Are those fruiting bodies all included in the analysis or are they cropped out? Are these the fruiting bodies outside the initial spot?

As you noticed, DZF1 and Δ mkapC - 0.5% do show bacterial activity outside the drop. However, fruiting bodies (seen as dark spots) are formed just in the case of DZF1. We do not know the reason why this phenomenon occurs, but it has been observed in all of our replica. These fruiting bodies are formed during development from cells migrating from within the drop, and are not an artefact of spilled bacteria outside the drop at the initial time. In order to analyze the images systematically, we cropped the fruiting bodies placed outside the drop, but recorded their presence under the variable "FB out", which was considered as a categorical variable in the FAMD analysis. This feature was also considered in the supplementary reaction norms. This has now been clarified in the Methods section.

4. For the following images reported in Fig 2a:

mkapC – from 1.0% to 2.5%

mkapA/C – 2.0% and 2.5%

pktC2/D1 – from 1.0% to 2.5%

All these grey-scale images show white regions mostly evident on the larger fruiting bodies, especially the larger ones, which. These white spots seem to be caused by the light reflection from the illumination system. The presence of these spots could have a strongly bias on the quantification of geometrical parameters (e.g. area, circularity) that could in turn affect the following multivariate analysis as well. To control for the data accuracy, we would suggest the authors to show area quantifications that were done using the automated system and manual quantification of a subset of fruiting bodies that show such white features. In case of significant discrepancy ordancy, the images should be taken again with an illumination system that comes from below the plate and provides a homogeneous light throughout the entire sample.

Indeed, images were taken with an illumination system in which light comes from below. We adapted this system to minimize light reflection. However, in the cases you mention, a white dot still appears in the top of the fruiting bodies. We do not know if this is a consequence of our setup or a biological property of those FBs. To avoid bias on the quantification we considered these fruiting bodies as fully darkened in the binarized images (see example below).

Original picture Binarized image Binarized corrected image

5. Line 75 - Strains that are used in the manuscript – Throughout the entire manuscript, there is not a single reference to the physiological differences of that the selected mutants strains have if compared relative to the reference DZF1 parental strain. We strongly suggest that the authors describe to make clear what are the mutants they selected and why they were selected them. Nayar VT. and Inouye S. have shown strong effects of these mutants on fruiting body development and sporulation compared to the reference strain DZF1. Given such associations, a better contextualization of the mutants and their biological relevance to the paper is desired needed.

This is indeed an important issue. Please see our response to your points 1 and 2. This information has now been included in the description of the biological material in the Methods section.

Also, In this study we focus on the potential for morphologic phenotypic changes to occur, but we do observe changes ranging from modest to dramatic upon which selection may subsequently act. These include changes in development time, spore maturation and, more indirectly, FB size and number. Moreover, it has been reported that mutants of the same regulatory network exhibit statistically significant phenotypic differences when comparing mutants with parental strain (DZF1), including differences in spore count and viability (Escalante et al., 2012). This suggests that the joint effect of the physical environmental and the genetic background on morphologic and developmental traits may also be of evolutionary relevance. We now mention this in the Discussion section.

6. Line 86 – Here the authors refer to a FB as “fully developed” when it completes the process of darkening. Are there any known quantifiable correlates to darkening (e.g. spore counts, size) or precedents of this definition that could be cited? If this definition is unique to this manuscript, this could be noted. This definition needs to be followed by a reference or a better explanation of what a fully developed FB is (spore counts? Size?). However, in case the authors are only presenting their definition of what a fully developed FB is (which is totally legit), I would suggest to be more direct and clear.

During FB development, some cells differentiate into spores while vegetative ones continue moving toward immature aggregates. Conventionally, full development and darkening of these aggregates is considered to occur at 72-96 h, when FBs attain a stable shape, size and darkness (e.g. Escalante et al., 2012 and O'Connor & Zusman, 1991). We did not consider the spore count and are not aware of how it precisely relates to darkness. This association is conventionally accepted in the work we have revised, and we clarify

this definition based on phenotype and include the two references where full development is considered at complete darkening at 72-96 h (Escalante et al., 2012 and O'Connor & Zusman, 1991).

7. Line 96 – (FB out) it is hard to understand how the quantification was done here given the difficulties of tracking the spot margin after 96h. Was the spot margin quantified right after the culture dried at 0 h? How? The spot size can be very different after 96h on TPM plates.

The drop is seen as a flat drop without any roughness at time 0 h, right after the culture dries. We visually established the edge and measured the drop diameter at this time using FIJI (ImageJ). From this moment, cells begin to move and develop FB, but the spot diameter does not change (see example below). Only FBs within this diameter were considered for the FB count. “FB out” is a categorical variable with value equal to 1 if matured FBs are observed outside the drop. We have briefly clarified this in the Methods section.

*mkapAC 0.5% . 0 h (5184x3456 pixels)
Drop diameter = 6795.80 μ m*

*mkapAC 0.5% . 96 h (5184x3456 pixels)
Drop diameter = 6732.57 μ m*

8. Line 97 - (Maturation) see comment for Line 86. Moreover, in table S1, “Maturation” is defined as a binary population trait (0 and 1). Therefore, I would one expect to see “Maturation” and “Scars” to show symmetric but opposite patterns to one another, as seems to be the case for only DZF1? If not, why not? And the definition of “Scars” could be clearer -(transparent, immature fruiting bodies?) - clear and not mature fruiting bodies(?) - to be symmetrical to one another, which is true only for DZF1 (Fig S2).

Thank you. We have now noticed that not all the meanings of the phenotypic traits were clearly explained, but have corrected this in our revision. The “Maturation” variable with value equal to one refers to complete darkening of clearly defined aggregates, while “Scars” equal to one refers to those drops presenting marks or vestiges of undefined aggregates (see illustrative close up below). These variables do not necessary correlate, as maturation seems to be more related with agar concentration, conversely to scars formation, which seems to be associated with the genotype. Δ pktC2/ Δ pktD1 - 1.0% is an example of Maturation=0, Scars=0, while Δ mkapA/ Δ mkapC illustrates the case of Maturation=1, Scars=1. To clarify this, we have now modified these definitions in the methods section and added panels in Figure 1.

Scars = 1

Scars = 0

9. Line 97 – (Dev. time) Given the explanation in table S1, “Dev. Time” and “Maturation” seem to overlap. What distinguishes them is not clear from the definition that has been provided.

Micrographs were taken at 96 h when FBs have fully developed (completely matured) in most cases. However, some aggregates do not present complete darkening at 96 h (e.g. $\Delta mkapC2/\Delta mkapD1$ at 0.5 %), which we indicate with the “Maturation” variable. On the other hand, even in the cases where complete maturation occurred, developmental time differs among genotypes and agar concentrations. For example, in general, development seems to occur slower at 0.5% than another agar concentration. The quantification of these differences in development time is expressed in the variable “Dev Time”, in which values go from 1 (fast development) to 5 (slow development). Below, we compare development of $\Delta mkapA$ for two moments, 36 h and 96 h. By 96 h both have reached complete development, but development of $\Delta mkapA$ 1.0% is faster than $\Delta mkapA$ 2.5%, as aggregates are already mature at 36 h. We have now clarified these definitions in the methods section.

36 h 96 h

$\Delta mkapA$ 1.0%

$\Delta mkapA$ 2.5%

10. Line 128 – Given the molecular background, isn't the overlap of observation that $mkapC$ and $mkapA/C$ overlap is as important as the one which would point to just as significant as the absence of overlap between $mkapA/C$ and $mkapA$ in low agar concentration (0.5% to 1.5%)? Why the emphasis on the latter?. However, this aspect raises the question on the biological meaning of such similarities/dissimilarities that could be discussed.

*Thank you for highlighting this issue. It is indeed interesting that the double mutant resembles one of the single mutants, but not the other. This shows that the double mutation does not have an additive effect, which is expected from the complexity of the network in which these proteins are involved (Arias Del Angel, et al. 2018). At this point we cannot propose any precise mechanism behind these non-additive phenotypes, beyond stating that our results demonstrate the complex control of *M. xanthus* development. In the reviewed version we now mention this non-additivity and contrast the similarity with the $\Delta mkaC$ mutant against the dissimilarity with the $\Delta mkaA$ mutant (first paragraph in the results section).*

11. Line 130 – The location of the DZF1 ellipse is interesting and could be discussed more. The relative position of DZF1 seems to change drastically from 0.5% to any the other agar concentrations. The position that the DZF1 ellipse occupies at higher agar concentrations doesn't vary as dramatically as the authors seem to suggest reported. However, it is striking that a similar trend, yet almost symmetrical to the one of DZF1, is shown by *pktC2/D1*. As for the previous comment, the biological relevance of this trend could be addressed expanded in the discussion.

Yes, the relative location of the DZF1 with respect to other genotypes is very interesting and has led us to speculate about the potential canalization or domestication of the strain. Indeed, other work has suggested that standard laboratory strains exhibit canalization (low plasticity) in important traits (Palková, 2004). This could be the case for DZF1, but at this point we have too little evidence to even suggest it. It would be necessary to compare it with wild strains -not mutants derived from the same strain- in order to further test this idea. In the current version we prefer to just say that "...plasticity is often studied in lab-adapted strains, and at least in this work, the parental lab-strain is not representative of the plastic responses for the rest of the genotypes (figures 1 and 2)."

*Regarding the position of the DZF1 ellipse in the 0.5 %, it does seem that this agar % affects the phenotypes so strongly that it surpasses the genotype arrangements maintained in all other agar %. We describe this in the results and also mention that, apart from 0.5 %, DZF1 relative location remains almost constant (first paragraph in the results section). As for the location of the *pktC2/D1* ellipse, it is very difficult to advance a potential explanation at this point, so we have decided not to comment it further in the main text.*

12. Line 148 – The authors mentions "Shape" but it is not clear to what parameter/s they are referring to in Figure 3. If they are referring to "Circularity", it would help to they should be consistent with the nomenclature present in the figure. Also, we suggest using 'lineal' in the same sentence, the author use the adjective 'lineal' instead of linear? If so, linear would be a more appropriate term in this case. Thus, when the author use the adjective linear in the next sentence when commenting on the trend of the trait "Area" the comparison will result to be clearer. for consistency with the next sentence.

Thank you for the observations. Indeed, we were referring to "Circularity" instead of "Shape" and we also meant to say "linear". We have corrected these mistakes.

13. One very surprising result piece of data that I have found particularly surprising is the reported relationship between number of fruiting bodies in the light of the variation of the

“Area” and the “area” parameter. Interestingly, DZF1 shows an almost steady increase growth in the number of FBs as agar concentration increases. Yet, especially when compared to the mutant strains, the area is not increasing nor decreasing without any corresponding change in the average area of individual FBs. This observation seems quite unexpected. Given the same number of cells (cell density is kept consistent in all experimental conditions), DZF1 forms up to 1000 FBs and yet average the FB area is barely decreased if not at all. Moreover, the measurement “Distance” parameter is also behaving unexpectedly. One expects that With an increase in FBs number and an with relatively constant “Area” should cause that doesn’t show differences, the variable “Distance” variable to should decrease - or at least not increase at all — yet this does not appear to be the case as the current plot doesn’t show (for DZF1). We think In case these quantifications will should be rechecked be confirmed by the authors. If they are confirmed as accurate, they should be discussed with possible explanations due to their surprising character, as they developed further within the paper discussion. In fact, such observations may be of interest to the study of the development of multicellular structures in a changing environments.

Thank you for this observation. We were also puzzled by this result. We have revised all the calculations and graphs and the results hold. The distance is calculated as the average euclidean distance among all the fruiting bodies in the population. Our explanation for this surprising behaviour of the variable “Distance” is that the FBs are not evenly distributed on the surface. Overall, they sometimes form clusters, rings or worm-like structures, which can affect the behaviour of the “Distance” variable. In the specific case you note, DZF1 at 2.5 % tend to accumulate FBs in the edge (ring), probably augmenting the distance because the FBs in the ring increase the distance to those in the colony slightly more than they decrease the distance to other FBs in the ring. Indeed, the median in the DZF1 “Distance” variable ranges from 2.75 to 3.0, while the number of FBs almost triples. We think that given the heterogeneous location of FBs, the “Distance” variable might exhibit slight but significant changes in unexpected directions, especially in the extreme substrate conditions, and we now mention this in the Methods section (subsection measurement of phenotypic traits).

Additional points:

14. Line 36-38 - The reference from Sultas S. 2015 (ref #1 in the paper), could go at the end of the sentence given its relevance to all expressed concepts. We suggest to adding more experimental paper references to the first part of the introduction (e.g. Monteiro et al 2013, Matsumoto et al 2013, Mayakawa et al 2010, Gilbert SF 2016) and not limit the material references source almost to reviews only.

Here few suggestions: - Monteiro et al 2013, Matsumoto et al 2013, Mayakawa et al 2010, Gilbert SF 2016,

Thank you for the references. They are indeed of major importance and we have included original sources.

15. Line 44 – I would strongly recommend the authors to cite West-Eberhard MJ’s book “Developmental Plasticity and Evolution. 2005” since this book (and other publications

from the same author during the same years) brought to debate and research more extensively the concept the authors express in this paragraph.

Thank you for the suggestion. We have included it in the main text.

16. Line 46 – the authors write “The origin of multicellularity is a major evolutionary transition.” I would be more specific adding that it is a “[...] major evolutionary transition in organismal development”.

OK, we have changed the sentence according to your suggestion.

17. Line 52 – “Importantly, physical processes and mechanical forces associated...”. I would suggest to rephrase this sentence. Right now, it is not clear why to clarify what distinction between mechanical forces are distinguished and not considered and physical processes is intended (or are they?).

Thank you. We have rephrased the sentence using only “physical processes”.

18. Line 93 – The authors refer to ImageJ v2.0.0 in the text but and cite FIJI (ref.# 16 in the manuscript).

FIJI (Fiji Is Just ImageJ) is a distribution of ImageJ including some specific plugins. We left both names for a better reference.

19. Line 108 – It seems would find more appropriate to refer to Sup. Table 2 and not rather than Sup. Table 1. Scree plots and a plot showing the contribution per dimensions would greatly help the reader in accessing the information present in Sup. Table 2.

Thank you for the observation. We have now corrected this mistake and included the suggested graphic summary.

20. Line 113 – Do all the measured traits have a symmetric distribution around the mean? If some are skewed, it may would be better to use medians instead.

Thank you for the suggestion. We have now used the medians and the results stayed the same. We updated the figure and associated tables considering the median instead of the mean.

21. Figures - Color codes in figures could be chosen to be more “color-blind friendly”. Specifically, Fig. 2b and Fig. 2c are very hard to navigate if the reader is affected by deuteranopia (same for Sup Fig 1). The authors may find useful the software Color Oracle (colororacle.org) to help testing the images for color- blind readers. In addition, 20. since the author are using ggplot2 package in R to generate images, they can make use of the `colorblind_pal()` function: <https://rdr.io/cran/ggthemes/man/colorblind.html>

Thank you, we checked the colors we used in the first version and changed those which represented as difficulties for color-blind readers.

22. Line 183 – It would be interesting to visualize such data in a representative panel (no quantifications needed).

Yes, we have now included this figure in the Supplementary material (no dark points corresponding to mature FBs are observed).

23. Fig 3 – We suggest reporting in this case, an inferential error (e.g. standard error or CI) should be preferred to rather than the standard deviation error (descriptive error).

Thank you for the suggestion. We have updated the figure using the standard error instead the deviation error. The general observations from the results stayed the same. Overall, when we considered the median and the standard error (regarding point 20), the error bars were considerably shrank.

24. Fig S1 – Legend text explaining graph features, e.g. the meaning of colors, is missing.

Thank you, we have explained graph features this in all figure legends.

25. Fig S2 – No explanation of missing error bars. exp.

Thank you for the observation. We have completed the missing information.

26. Table S1 – “Circularity” should be noted as dimensionless.

Thank you for noticing this error. We have corrected it.

References

- Andac, T., Weigmann, P., Velu, S. K., Pinçe, E., Volpe, G., Volpe, G., & Callegari, A. (2019). Active matter alters the growth dynamics of coffee rings. *Soft matter*.
- Arias Del Angel, J. A., Escalante, A. E., Martínez-Castilla, L. P., & Benítez, M. (2018). Cell-fate determination in *Myxococcus xanthus* development: Network dynamics and novel predictions. *Development, growth & differentiation*, 60(2), 121-129.
- Arias Del Angel, J. A., Escalante, A. E., Martínez-Castilla, L. P., & Benítez, M. (2017). An Evo-Devo Perspective on Multicellular Development of Myxobacteria. *Journal of Experimental Zoology Part B: Molecular and Developmental Evolution*, 328(1-2), 165-178.
- Escalante, A. E., Inouye, S., & Travisano, M. (2012). A spectrum of pleiotropic consequences in development due to changes in a regulatory pathway. *PloS one*, 7(8), e43413.
- Nariya, H., & Inouye, S. (2006). A protein Ser/Thr kinase cascade negatively regulates the DNA-binding activity of MrpC, a smaller form of which may be necessary for the *Myxococcus xanthus* development. *Molecular microbiology*, 60(5), 1205-1217.
- Nariya, H., & Inouye, S. (2005). Modulating factors for the Pkn4 kinase cascade in regulating 6-phosphofructokinase in *Myxococcus xanthus*. *Molecular microbiology*, 56(5), 1314-1328.
- Nariya, H., & Inouye, S. (2003). An effective sporulation of *Myxococcus xanthus* requires glycogen consumption via Pkn4-activated 6-phosphofructokinase. *Molecular microbiology*, 49(2), 517-528.
- Nariya, H., & Inouye, S. (2002). Activation of 6-phosphofructokinase via phosphorylation by Pkn4, a protein Ser/Thr kinase of *Myxococcus xanthus*. *Molecular microbiology*, 46(5), 1353-1366.
- O'Connor, K. A., & Zusman, D. R. (1991). Development in *Myxococcus xanthus* involves differentiation into two cell types, peripheral rods and spores. *Journal of bacteriology*, 173(11), 3318-3333.
- Palková, Z. (2004). Multicellular microorganisms: laboratory versus nature. *EMBO reports*, 5(5), 470-476.
- Shi, W., & Zusman, D. R. (1993). The two motility systems of *Myxococcus xanthus* show different selective advantages on various surfaces. *Proceedings of the National Academy of Sciences*, 90(8), 3378-3382.